# Characterization of Vitamin D Status in Older Persons with Cognitive Impairment

**DOI:** 10.3390/nu14061142

**Published:** 2022-03-08

**Authors:** Beatrice Arosio, Paolo Dionigi Rossi, Evelyn Ferri, Matteo Cesari, Giovanni Vitale

**Affiliations:** 1Department of Clinical Sciences and Community Health, University of Milan, Via Pace 9, 20122 Milan, Italy; matteo.cesari@unimi.it; 2Geriatric Unit, Fondazione IRCCS Ca’ Granda Ospedale Maggiore Policlinico, Via Pace 9, 20122 Milan, Italy; paolo.rossi@policlinico.mi.it (P.D.R.); evelyn.ferri@guest.unimi.it (E.F.); 3Geriatric Unit, IRCCS Istituti Clinici Scientifici Maugeri, Via Camaldoli 64, 20138 Milan, Italy; 4Department of Medical Biotechnology and Translational Medicine, University of Milan, Via Vanvitelli 32, 20133 Milan, Italy; giovanni.vitale@unimi.it; 5Istituto Auxologico Italiano IRCCS, Laboratory of Geriatric and Oncologic Neuroendocrinology Research, Via Zucchi 18, 20095 Cusano Milanino, Italy

**Keywords:** aging, vitamin D, cognitive impairment, frailty, dementia

## Abstract

Vitamin D exerts a role in the maintenance of cognitive abilities and in frailty. Although several studies evaluated the interactions between vitamin D and cognitive impairment, results were conflicting. In a cohort of community-dwelling older persons, we described the association between vitamin D levels and cognitive decline and all-cause dementia evaluating frailty’s contribution. Our cohort included 509 adults, aged 64–92 years: 176 patients with mild cognitive impairment (MCI), 59 with Alzheimer’s Disease (AD), 26 with idiopathic Normal Pressure Hydrocephalus (iNPH), 133 with mixed dementia (MD) and 115 without cognitive decline. Frailty was measured by frailty index, and serum 25-hydroxyvitamin D concentrations through electrochemiluminescence immunoassays. We found a significant association between vitamin D levels and Mini Mental State Examination independently of cognitive impairment, age, sex and frailty. The patients with dementia (AD and MD) showed the lowest vitamin D levels, while MCI patients showed higher levels than the other groups. The most severe deficiency was observed in MD patients, the most aged as well as cognitively and functionally impaired. In conclusion, in our community-dwelling older persons investigated for a suspected cognitive impairment, we observed an association between vitamin D levels and cognitive decline, regardless of the frailty status.

## 1. Introduction

Aging induces changes in all body apparatus, including the remodeling of the endocrine system [1]. This remodeling involves the parathyroid glands altering the production of the parathyroid hormone and thus of the secosteroid pro-hormone vitamin D [2].

In general, the consequences of these changes are manifold and include the reduction of bone and skeletal muscle mass, the reduction in muscle strength and the increase in adipose tissue driving the “fragilization” in older persons [3,4,5,6]. However, the effects of vitamin D are not confined to the musculoskeletal system [7]. Vitamin D appears to have a role in the development of the brain and in the maintenance of cognitive abilities [8,9].

Interestingly, a recent study reported that in a new mouse model the vitamin D receptor (VDR) is widely distributed into the brain cells [10,11] expressing the enzyme that converts the inactive vitamin D form into the active one [12]. This pathway makes the brain able to synthesize, catabolize and receive vitamin D, which regulates many cellular processes in neurons and microglial cells [12]. In this regard, we have previously described an association between the genetic profile of the VDR gene and the cognitive performances in a cohort of centenarians [13].

From a functional point of view, vitamin D is synthesized by neurons and microglia, that use the active hormone to regulate cell proliferation, differentiation and survival [14]. Vitamin D affects essential processes for the brain development in the embryo, e.g., synaptic plasticity as well as cytoskeleton maintenance [15,16]. Moreover, vitamin D exerts an anti-inflammatory and neuroprotective action within the brain through the modulation of the immune response interacting with the innate and adaptive immune system [17]. Vitamin D exerts its anti-inflammatory activity also inhibiting the nitric oxide synthesis and reducing the oxidative burden within neurons and microglia [18,19].

For all these reasons, deficiency of vitamin D has been proposed to promote cognitive dysfunction and brain atrophy. In fact, the correlation between vitamin D and dementia, in particular Alzheimer’s disease (AD), has been explained by impressive studies in vitro and in animal models [20]. To date, the findings obtained in humans are controversial [20].

Indeed, there is certainty that vitamin D has a role in normal brain function and that low vitamin D levels can occur in patients with dementia. In this regard, in some prospective and cross-sectional studies, the lower concentrations and the deficiency of vitamin D were associated to an increased risk of developing AD [21,22,23,24,25,26].

At the same time, in persons from the Original and Offspring Framingham Heart Study [27], low vitamin D levels were not significantly associated with the onset of dementia or AD, while vitamin D deficiency was associated with poor domain-specific cognitive performance (e.g., executive function, processing speed and visual–perceptual skills) and small hippocampal volume [27].

It is to note that these associations may be influenced by socio-economic status, clinical comorbidity, and past medical history as demonstrated in older persons belonging to a population-based study from Korea [28]. In this study, the authors showed that vitamin D deficiency was not significantly associated with a decline in cognitive function compared to vitamin D sufficiency after adjusting by the above reported factors [28]. However, the authors did not consider vitamin D supplementation in this population.

Interestingly, low vitamin D levels have been associated with lower functioning and greater cognitive decline over time in the cohorts of the Health Aging and Body Composition Study [29], of the Longitudinal Aging Study Amsterdam and US Cardiovascular Health Study [30] and of the ESTHER study [31,32]. Recently, vitamin D deficiency was associated with a modest reduction of memory function in absence of neurodegeneration and vascular changes in older persons with normal neurological status and without history of stroke and dementia belonging to the Austrian Stroke Prevention Study (ASPS) and the Austrian Stroke Prevention Family Study (ASPS-Fam) [33].

The reason for these discrepancies could be a lack in consensus on the cut-off values that define circulating vitamin D sufficiency, deficiency and insufficiency, as well as to the different assay methods used in the different studies [34].

Thus, vitamin D’s reliability as a serum biomarker in cognitive decline and dementia is yet considered a debated issue [34].

In this study, we aimed to characterize the vitamin D status in a cohort consisting of community-dwelling older persons investigated for a suspected cognitive decline. We also aimed to describe the association between vitamin D levels and cognitive decline as well as all-cause dementia evaluating the possible contribution of frailty.

## 2. Materials and Methods

### 2.1. Study Design

For this study, we considered 509 community-dwelling older adults (64–92 years) who consecutively attended a first geriatric visit to the geriatric unit of the Fondazione IRCCS Ca’ Granda Ospedale Maggiore Policlinico (Milan, Italy) from March 2009 to February 2018. All participants were admitted to the geriatric unit for the investigation of a suspect cognitive decline and their data were recorded in “Registro di Raccolta Dati della Unità di Geriatria” (REGE 2.0).

All participants underwent a multidimensional geriatric assessment during which information about their medical history and cognitive and functional statuses were recorded.

The cognitive status is assessed by means of a modified version of the Mini Mental State Examination (MMSE) [35] and a battery of neuropsychological tests (i.e., Trail Making Test, Verbal Fluency Test, Digit Span Forward and Backward Tests, Verbal Learning Tests, Token Test, Rey’s Figure Copy and Delayed Recall, Raven Coloured Progressive Matrices). The presence of depression was evaluated by using the Geriatric Depression Scale (GDS). The functional status was assessed by means of the Activity of Daily Living (ADL) and the Instrumental Activity of Daily Living (IADL) scales. Two different cut-offs were applied to the latter scale for men and women [36].

All persons taking vitamin D supplementation were excluded from the study. Among the examined subjects, 176 patients met the criteria of mild cognitive impairment (MCI) outlined by Petersen [37], 59 were diagnosed with Alzheimer’s Disease (AD) [38], 26 were diagnosed with idiopathic Normal Pressure Hydrocephalus (iNPH) according to the International Guidelines [39] and 133 were identified as mixed dementia (MD) patients characterized by both neurodegenerative and vascular findings [40]. A total of 115 subjects (controls) were diagnosed without cognitive decline (i.e., MMSE score ≥ 24, NPS test negative for dementia, no neurological or psychiatric disorders).

All subjects consented personally or through their legal guardian (if the person is mentally incapable of independently making a decision) to the use of their clinical and biological data for appropriately anonymized research purposes.

The study protocol received approval from the local ethics committee (REGE 2.0, protocol number 1696, 4 June 2021).

### 2.2. Frailty Index

The frailty was defined in relation to the accumulation of deficits in the Frailty Index (FI). We scored each individual’s FI by counting the number of deficits and dividing this count by the total number of deficits considered. The items comprised biochemical parameters, signs, disabilities and diseases (Table 1).

Each item of the FI was scored as ‘0’ (absence of the deficit) or ‘1’ (presence of the deficit). The FI was calculated as the ratio between the number of health deficits of the subject and the total number of health deficits considered for its computation (*n* = 44) (Table 1).

Although the FI is determined on a continuous scale, the scale can be subdivided for comparisons of different health states. Our cohort according to literature [41,42] included persons less fit (0.03 < FI ≤ 0.10), least fit (0.10 < FI ≤ 0.21), frail (0.21 < FI < 0.45) and most frail (FI ≥ 0.45).

### 2.3. Vitamin D

Serum 25-hydroxyvitamin D (vitamin D) concentrations were measured through Cobas electrochemiluminescence immunoassays (ECLIA) on the Modular E170 analyzer (Roche Diagnostics, Mannheim, Germany).

This method was standardized against LC-MS/MS [43] which, in turn, was standardized against the NIST standard [44]. The performance of this assay was validated using Elecsys reagents, human sera and controls, performed according to the protocol EP5-A2 of the Clinical and Laboratory Standards Institute (CLSI).

The functional sensitivity of the assay was 4.01 ng/mL (CV 18.5%). The assay specificity (reflecting by the percentage of cross reactivity with other metabolites) was 100% for 25(OH)D_3_, 92% for 25(OH)D_2_, and 91% for the C_3_ epimer of 25(OH)D_3_.

A vitamin D level of 30 ng/mL or higher was considered adequate, whereas a concentration from 12 to 30 ng/mL was considered a vitamin D insufficiency and below 12 ng/mL a vitamin D deficiency.

### 2.4. Statistical Analysis

An SPSS statistical package was used to conduct the statistical analyses (SPSS 27, Chicago, IL, USA).

Age, education, MMSE, ADL, IADL and GDS were expressed as mean (standard deviation, SD). FI and vitamin D concentrations were expressed as median (interquartile range, IQR).

The parametric variables were analyzed using the Analysis of Variance (ANOVA) and the Bonferroni post-hoc test, whereas the non-parametric variables were analyzed using the Kruskal–Wallis test and the Mann–Whitney U test.

Since vitamin D was not normally distributed, the data were log-transformed for regression analyses. The general linear model was applied to compare log-transformed vitamin D values between men and women, including age as covariate. Both univariate and multivariate linear regression analyses adjusted by confounding factors (i.e., sex, age, FI and diagnoses) were used to investigate the association between log-transformed vitamin D values and MMSE.

A *p* < 0.05 was considered as the threshold for statistical significance.

## 3. Results

The analyzed data are the baseline information recorded at recruitment. The characteristics of the cohort are described in Table 2. Briefly, the mean age of the REGE cohort was 79 (SD 5) years and the mean value of MMSE was 24 (SD 5). The median FI value was 0.27 (from 0.04 to 0.62). The median level of vitamin D was 13 ng/mL (from 3 ng/mL to 58.9 ng/mL) (Table 2). The percentage of women was greater than that of men (60.5% and 39.5%, respectively).

The univariate linear regression analysis showed a positive association between vitamin D levels and MMSE (R^2^ = 0.06, β = 0.24, *p* < 0.001) (Figure 1). Adjusting for age, sex, FI and diagnoses, the multivariate linear regression analysis confirmed the positive association between vitamin D levels and MMSE (R^2^ = 0.20, β = 0.16, *p* < 0.001).

A slight difference was found in vitamin D levels between women and men (12, SD 7–20 and 15, SD 9–21, respectively), adjusting for age (*p* = 0.04).

Categorizing the subjects on diagnoses, MD patients were significantly older than MCI patients (*p* < 0.001).

As expected, we observed differences in education, MMSE, ADL, IADL and GDS among the groups (Table 3). In particular, MD patients showed a lower educational level than controls and MCI patients (*p* < 0.001). MMSE was lower in patients with AD, iNPH and MD compared to controls and MCI patients (*p* < 0.001). The groups more functionally compromised were iNPH (ADL: *p* < 0.001 vs. controls, MCI and AD; IADL: *p* < 0.001 vs. controls, MCI and *p* = 0.03 vs. AD) and MD (ADL: *p* < 0.001 vs. controls, MCI and *p* = 0.007 vs. AD; IADL: *p* < 0.001 vs. controls and MCI; *p* = 0.01 vs. AD) (Table 3). Finally, the less depressed were AD (*p* = 0.008 vs. controls and *p* < 0.001 vs. MD) and MCI patients (*p* = 0.003 vs. MD). iNPH and MD patients were frailer than controls, AD and MCI patients (*p* < 0.001) (Table 3).

Regarding vitamin D, both AD and MD patients were characterized by lower levels of vitamin D compared to MCI patients (*p* = 0.02 and *p* < 0.001, respectively). Interestingly, MCI showed higher levels than the other groups (*p* < 0.001 vs. MD, *p* = 0.02 vs. AD, *p* = 0.03 vs. iNPH and *p* = 0.009 vs. controls) (Table 3).

## 4. Discussion

Vitamin D is a secosteroid hormone that plays a key role in various physiological processes, ranging from the modulation of the immune response to the regulation of brain development. Vitamin D exerts anti-inflammatory and neuroprotective activities in the brain acting on the synthesis of proinflammatory cytokines and reducing oxidative stress load. Moreover, Vitamin D has a role in the clearance of amyloid plaques by the immune cell [20]. Therefore, there is growing preclinical evidence about the capability of vitamin D in the prevention of amyloid accumulation and thus of cognitive decline [45].

Although several studies evaluated the interactions between vitamin D levels and cognitive impairment, results from literature are conflicting, and there is no clear consensus about this association.

This study, conducted in a cohort of community-resident older persons not taking vitamin D supplementation, provides several interesting hints.

First, a significant association was found between vitamin D and MMSE in the overall subjects, independently of the presence/absence of the cognitive impairment. The association was confirmed also adjusting by age, sex, frailty and diagnoses of cognitive impairment. The patients affected by dementia (AD and MD) showed lower levels of vitamin D, while MCI showed higher levels compared to the other groups.

Several studies have investigated the potential relationship between serum vitamin D and cognitive function and obtained conflicting results. Most cross-sectional studies concern the association between vitamin D deficiency and increased AD risk [20,21,24,25,26,27,46,47,48,49]. Likewise, the prospective studies having investigated cognitive impairment in older persons over time have furnished confusing results [24,29,30,31,32,46,50,51,52,53,54,55,56].

Methodological issues could explain some of these discrepancies and, in particular, the lack of consensus in defining the concentrations that discriminate the deficiency. As expected, the majority of our older persons showed a vitamin D deficiency (redefined in 1998 as a blood level of 25(OH)D < 20 ng/mL) [57].

Another methodological issue could be related to the presence of depression, which can interfere with the MMSE values. As expected, in our cohort, AD patients showed the lowest GDS score [58,59], while the other groups had a comparable GDS values similarly affecting MMSE.

It is important to note that most studies consider only age and sex as potential confounders, whereas few studies adjusted the analyses also by frailty [60].

Recently, deficient levels of vitamin D were identified as a key factor contributing to the development of frailty in very older persons [4]. Therefore, interventions to prevent hypovitaminosis D have been proposed to slow down the process of “fragilization” that occurs during aging [61,62].

Moreover, frailty is correlated with a decline in cognitive function [63,64] and high frailty scores are associated with a faster rate of decline in all cognitive domains [63,65]. For these reasons, we have decided to adjust the results of frailty using the FI and we have discovered an association between vitamin D levels and cognitive decline, regardless of the person’s “fragilization”.

As expected, our results showed slightly lower levels of vitamin D in women than men, after adjusting for age [66].

Interestingly, we observed the lowest vitamin D levels in MD patients, the most impaired from both cognitive and functional point of view, even compared to AD patients. As described by Perrotte et al. [60], no difference was observed between the mild and/or moderate AD patients and controls of our cohort.

It should be noted that vitamin D triggers various neural pathways, potentially exerting a protective function against neurodegenerative processes and justifying its relationship with AD [20].

In fact, our results seem to suggest a role of vitamin D mainly in patients with MD characterized by both neurodegenerative and vascular findings. Interestingly, it has been proven that vitamin D exerts a variety of favorable effects on endothelial dysfunction [67]. This could explain its protective role in a form of dementia characterized by vascular etiopathogenesis [68].

One limitation of this study could be that our MD patients were older, less literate and frailer compared to the other groups. To compensate, our results were adjusted by age and FI.

In conclusion, in this study, we observed an association between vitamin D levels and cognitive decline, regardless of frailty status, age and gender, in a cohort of community-dwelling older persons not taking vitamin D supplementation investigated for a suspected cognitive decline.

## Figures and Tables

**Figure 1 nutrients-14-01142-f001:**
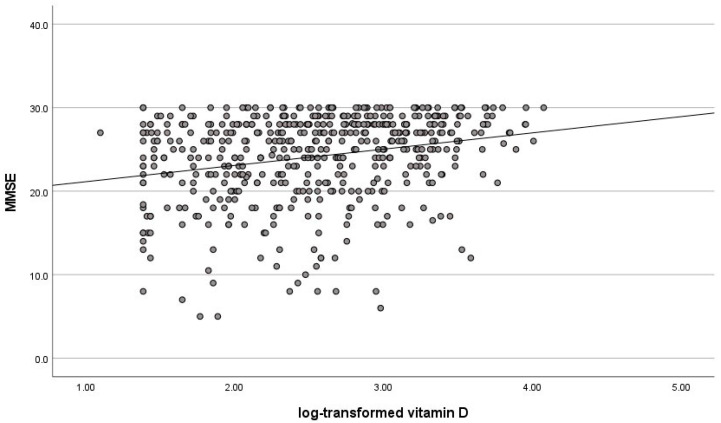
Univariate linear regression analysis between log-transformed Vitamin D and MMSE score in all subjects.

**Table 1 nutrients-14-01142-t001:** List of the 44 biochemical and health deficits included in the FI.

Deficit	
Biochemical Parameters	
Cholesterol	≤200 mg/dL
CRP	<0.5 mg/dL
Vitamin B12	191–663 ng/L
Folate	4.6–18.7 µg/L
TSH	0.28–4.30 mIU/L
Signs	
Pain	
Bowel incontinence	
Sleep disorders	
BMI	21–30 kg/m^2^
Edema	
Tremor	
Disabilities	
Mobility impairment	
ADL-disability in self-feeding	
ADL-disability in dressing	
ADL-disability in bathing	
ADL-disability in transferring	
ADL-disability in toileting	
ADL-incontinence	
IADL-disability in using telephone	
IADL-disability in shopping	
IADL-disability in food preparation	
IADL-disability in housekeeping	
IADL-disability in doing laundry	
IADL-disability in travelling by car or public transportation	
IADL-disability in medication use	
IADL-disability in handling finances	
Diseases	
Hypertension	
Diabetes	
Congestive heart failure	
Coronary heart disease	
Cardiac arrhythmia	
Chronic obstructive pulmonary disease	
Decreased visual acuity	
Hearing loss	
Osteoarthritis	
Vascular endothelial abnormalities	
Chronic renal insufficiency	
Hepatopathy	
Depression	
Cerebrovascular disease	
Cancer	
Osteoporosis	
Anemia	
Diverticulosis	

CRP: C-reactive Protein; TSH: Thyroid-Stimulating Hormone; BMI: Body Mass Index; ADL: Activity of Daily Living; IADL: Instrumental Activity of Daily Living.

**Table 2 nutrients-14-01142-t002:** Characteristics and vitamin D levels of the overall cohort.

	Mean (SD)	Median (IQR)
Age	79 (5)	
Education (years)	9.4 (4.6)	
MMSE score	24 (5)	
ADL score	5.0 (1.3)	
IADL score	5.1 (2.4)	
GDS	11 (6)	
FI		0.27 (0.21–0.36)
Vitamin D (ng/mL)		13 (8–21)

Education was reported as years of schooling; MMSE: Mini Mental State Examination; ADL: Activity of Daily Living; IADL: Instrumental Activity of Daily Living; GDS: Geriatric Depression Scale; FI: Frailty Index.

**Table 3 nutrients-14-01142-t003:** Characteristics and vitamin D levels of the subjects categorized on diagnosis.

	Controls (*n* 115)	AD (*n* 59)	MCI (*n* 176)	iNPH (*n* 26)	MD (*n* 133)	*p*
Age	80 (6)	78 (5)	78 (5)	81 (6)	80 (5)	<0.001
Education (years)	10.1 (4.5)	8.7 (4.8)	10.3 (4.6)	10.5 (4.4)	7.9 (4.0)	<0.001
MMSE	27 (4)	21 (5)	27 (2)	21 (5)	20 (5)	<0.001
ADL	5.2 (1.0)	5.1 (1.3)	5.4 (0.8)	4.0 (1.8)	4.5 (1.5)	<0.001
IADL	5.9 (2.3)	4.9 (2.6)	5.9 (1.9)	3.3 (1.9)	3.8 (2.3)	<0.001
GDS	12.5 (6.6)	7.8 (4.6)	10.5 (5.7)	12.4 (4.8)	14.4 (7.0)	<0.001
FI	0.24 (0.17–0.32)	0.25 (0.19–0.33)	0.25 (0.19–0.31)	0.34 (0.26–0.41)	0.34 (0.24–0.42)	<0.001
Vitamin D (ng/mL)	12 (7–22)	12 (8–20)	17 (10–25)	13 (5–19)	10 (6–16)	<0.001

Education was reported as years of schooling; MMSE: Mini Mental State Examination; ADL: Activity of Daily Living; IADL: Instrumental Activity of Daily Living; GDS: Geriatric Depression Scale; FI: Frailty Index. Controls: subjects without cognitive decline; AD: Alzheimer’s disease; MCI: Mild Cognitive Impairment; iNPH: idiopathic Normal Pressure Hydrocephalus; MD: Mixed Dementia.

## Data Availability

The data presented in this study are available on request from the corresponding author.

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
