# Peer review of "Characterization of Vitamin D Status in Older Persons with Cognitive Impairment"

_nutrients, 2022, doi:10.3390/nu14061142_

Round 1

Reviewer 1 Report

Overall the study is well-conceived and the manuscript well-written. The authors present the results of original research and make a valuable contribution to knowledge and understanding of vitamin D status in older person with cognitive impairment.

However, in my opinion the paper has some shortcomings in regards some paragraphs, for example in Results Authors reported that 122 patients were taking vitamin D supplementation, but they did not indicate the levels of vitamin D supplementation and the subjects of this group. I suggest considering to exclude them from analysis or to consider them as a separate group.

Moreover, I don’t understand the method that Authors used to calculate the frailty index. There was a score? How subjects were classified?

Author Response

Reviewer 1

However, in my opinion the paper has some shortcomings in regards some paragraphs, for example in Results Authors reported that 122 patients were taking vitamin D supplementation, but they did not indicate the levels of vitamin D supplementation and the subjects of this group. I suggest considering to exclude them from analysis or to consider them as a separate group.

We thank the Reviewer and, as suggested, in the new version of the paper we have excluded the persons taking vitamin D supplementation from the analyses.

Moreover, I don’t understand the method that Authors used to calculate the frailty index. There was a score? How subjects were classified?

As suggested by the Reviewer, we have implemented the “materials and methods” paragraph with the detailed description of the FI calculation (page 5 lines 24-27 and 31-34). 

Reviewer 2 Report

This article aims to find an association between Vitamin D and cognitive impairment in frail and not frail older patients. This article could be interesting although several information is not available:

Major

The statistics should be checked again, the results and overall the clinical characteristics seems too similar to be significantly different. Check carefully Table 3 and 4.

The authors must insert the correlation coefficient

A graph with the confidence interval and another with the correlation is mandatory.

It will be really interesting to know if there re difference among sex, adjusted for the age.

Minor

Introduction:

-The authors must explain more extensively the rationale in measuring Vitamin D in cognitive impairment. Some sources are cited, but not explained.

- The following sentence is not clear, please reformulate it: " the key roles of vitamin D in the brain are the maintenance of the synaptic plasticity, the neurotransmission in dopaminergic neural circuits, and the reduction of the synthesis of pro-inflammatory cytokines and oxidative stress into the brain". What are the key roles?

Methods:

-Vitamin D: The authors must insert the extended name of the analyzed vitamin D. "Serum-25-OH-Vitamin D " is not correct. 

Tables and results:

- The authors must insert the measurement unit of Vitamin D

- The results on the vitamin d correlation with supplemented subject, should be available. The unavailability of the data indicating the period of supplementation should be stated as a limitation of the study. If the authors have this results, it must be showed.

Discussion:

Mild cognitive patients had the lowest Vitamin D, but they were older, less literate and frailed compared to the other group. This fact must be stated clear as a limitation of the study. For this reason, it is not recommended to insert in line that “we confirmed an association between vitamin D levels and cognitive decline, regardless of frailty status, age and gender, in a cohort of community-dwelling older persons investigated for a suspected cognitive decline.”.

Author Response

Reviewer 2

 Major

The statistics should be checked again, the results and overall the clinical characteristics seems too similar to be significantly different. Check carefully Table 3 and 4.

As suggested by the Reviewer, we have carefully checked the results of the statistical analyses.

It is to note that, accordingly by the criticism raised by the Reviewer 1, in the new version of the paper we have excluded the persons taking vitamin D supplementation from the analyses.

Thus, the comparisons between the groups of diagnosis have undergone some changes (Table 3).

A paradigmatic example of our statistical output on the vitamin D levels is reported below.

Kruskal-Wallis test

Null hypothesis

Test

Significance

Decision

1

The vitamin D distribution is the same among the categories of diagnosis.

Kruskal-Wallis

,000

Reject the null hypothesis

Pairwise comparisons

Test’s statistics

Standard error

Standard test’s statistics

Significance

Mod. Sign.

MD-iNPH

11,161

31,537

,354

,723

1,000

MD-AD

27,620

23,006

1,201

,230

1,000

MD-Controls

33,631

18,728

1,796

,073

,725

MD-MCI

79,984

16,898

4,733

,000

,000

iNPH-AD

16,459

34,620

,475

,634

1,000

iNPH-Controls

22,470

31,938

,704

,482

1,000

iNPH-MCI

68,823

30,901

2,227

,026

,259

AD-Controls

6,011

23,552

,255

,799

1,000

AD-MCI

-52,364

22,125

-2,367

,018

,179

Controls-MCI

-46,353

17,635

-2,628

,009

,086

The authors must insert the correlation coefficient.

We thank the Reviewer for the suggestion. In the new version, we have specified the correlation coefficient (R2) generated from both univariate and multivariate linear regression analyses adjusted by sex, age, FI and diagnoses (page 8 lines 12-15).    

A graph with the confidence interval and another with the correlation is mandatory.

As indicated by the Reviewer, in the “results” section we have added the graph with the univariate linear regression analysis between log-transformed Vitamin D and MMSE in our cohort (Fig. 1).

It will be really interesting to know if there are difference among sex, adjusted for the age.

As suggested, the difference in vitamin D levels between men and women was evaluated by means of the general linear model, adjusted for age (page 9 lines 16-17).

Minor

Introduction:

The authors must explain more extensively the rationale in measuring Vitamin D in cognitive impairment. Some sources are cited, but not explained.

The rationale in measuring vitamin D in cognitive impairment has been better described in the introduction of the paper (page 3 lines 28-29, 32-34 and page 4 lines 1-8). 

The following sentence is not clear, please reformulate it: " the key roles of vitamin D in the brain are the maintenance of the synaptic plasticity, the neurotransmission in dopaminergic neural circuits, and the reduction of the synthesis of pro-inflammatory cytokines and oxidative stress into the brain". What are the key roles?

We apologize for the low clarity of the sentence. In the revised version, we have reformulated it (page 3 lines 16-23).

Methods:

Vitamin D: The authors must insert the extended name of the analyzed vitamin D. "Serum-25-OH-Vitamin D " is not correct. 

In the new version, we have inserted the extended name of the analysed vitamin D as pointed out by the Reviewer (page 7 line 8).   

Tables and results:

The authors must insert the measurement unit of Vitamin D

In the tables and in the text, we have now specified the measurement unit of Vitamin D.

The results on the vitamin d correlation with supplemented subject, should be available. The unavailability of the data indicating the period of supplementation should be stated as a limitation of the study. If the authors have this results, it must be showed.

We agree with the Reviewer and accordingly to the criticism raised also by the Reviewer 1, in the new version of the paper we decided to exclude persons taking vitamin D supplementation.

Discussion:

Mild cognitive patients had the lowest Vitamin D, but they were older, less literate and frailed compared to the other group. This fact must be stated clear as a limitation of the study. For this reason, it is not recommended to insert in line that “we confirmed an association between vitamin D levels and cognitive decline, regardless of frailty status, age and gender, in a cohort of community-dwelling older persons investigated for a suspected cognitive decline.”

According to the Reviewer suggestion, we have clearly stated as a limit of the study the fact that the patients with mixed dementia (MD) were older and more frail than the other subjects (page 12 lines 8-9).

Anyway, we have adjusted all our analyses for age and frailty status (page 8 lines 13-15).

Reviewer 3 Report

The manuscript by Arosio et al., sought to investigate the association between vitamin D status and cognitive ability in community-dwelling elders above age 64 from Northern Italy.

Overall, the study could be considered, essentially, confirmatory research, the only novelty being that frailty was taken into account as a potential confounder.

The manuscript is quite readable although it is littered by inaccuracies. The background information provided in the introduction and the presentation of methods and data, in my opinion, are quite poor and the whole text needs a thorough review.

Major criticism

Introduction

Page 1, line 39. The literature regarding the precise localization of the vitamin D receptor in animal and human brain is quite discordant, and largely depends on the antibodies used in immunohistochemistry experiments (see a critical discussion in PMID: 33368246). The review cited in reference no. 10 (which in turn refers to PMID: 15589699) is out of date.

Page 2, line 54. The authors provided a list of studies that, they claim, do not support the low vitamin D/dementia association. However, the list includes PMID: 26890771 which, although did not find association between vitamin D and incident dementia, showed that low vitamin D levels were associated with poorer executive function; the study PMID: 32080146 was only cross-sectional and did not investigate for intake of vitamin D supplements; PMID: 31827432 was clearly underpowered and with too short a follow-up. These references are therefore inappropriately cited.

Methods

Page 2, lines 81-85. I have noticed a lack of detail regarding the recruitment of elders. How and when were the participants recruited into the study? Was it a random sample of all persons aged 64 or over, or a consecutive series? What was the participation rate?

Page 2, line 86-90. Since this is a study on dementia, more information should have been given about the psychometric scales adopted. For example, has the MMSE been adjusted for age and formal education according to PMID: 21284770? In addition, were the participants taking antidepressants? Could this have possibly led to interference with the accuracy of cognitive assessment, perhaps favouring lower depression scores?

Page 2, line 91. Why was the MMSE score threshold of 27 chosen to define the absence of dementia? While most studies usually adopt the ≤ 24 threshold for dementia, other studies adopt other thresholds (reviewed in PMID: 25740785).

Lines 93-94. The information provided by the authors about the study approval by the ethics committee, and the procedure for acquiring informed consent are undetailed and lacking documentary indications. This is even more serious since most participants were demented, for whom precise guidelines exist (see PMID: 33771131).

Page 3, lines 96-100. The description the frailty index calculation lacks details. If questions were asked directly to the elders, how demented or severely depressed participants who did not fully understand the questionnaire were managed? In addition, the instrumental activities of daily living scale (Table 1) were used to calculate the fragility index. But given that in Latin culture males rarely do housework, or do laundry or cooking, how the resulting biased responses to these items have been corrected?

Page 4, lines 118-120. Plasma vitamin D levels do not typically have a normal distribution but are somewhat skewed toward higher values (PMID: 1503066). How can the authors think of entering them into a linear regression model without prior transformation?

Results

Page 4, lines 129-130. If I understand correctly, education was not expressed as the number of years spent in school, but was categorized into five levels of unequal duration. It is not easy, in this case, to interpret the meaning of the corresponding regression coefficient.

Discussion

Page 6, lines 191-193. This sentence, at the beginning of the discussion is, word by word, identical to a sentence in the introduction (lines 45-48). While this is not a flaw in itself, it denotes, at least, poverty of language.

Minor remarks

The text is quite readable, but occasionally there are some misspellings or grammar problems, for instance:

Abstract (line 17) and page 2, line 88. The correct term is "Hydrocephalus" not "Hydrocephalous".

Line 21. Better “persons taking no supplements”.

Introduction. Line 63, better “enrolled in the study” than “belonging to the study”. Line 65, please correct “lack of consensus”. Line 68, please correct “is still considered a debatable issue”. Line 109, better replace “lacking levels” into “deficiency”

Author Response

Reviewer 3

Major criticism

Introduction

Page 1, line 39. The literature regarding the precise localization of the vitamin D receptor in animal and human brain is quite discordant, and largely depends on the antibodies used in immunohistochemistry experiments (see a critical discussion in PMID: 33368246). The review cited in reference no. 10 (which in turn refers to PMID: 15589699) is out of date.

As suggested by the Reviewer, the paragraph describing the vitamin D receptor has been better revised in the new version of the paper (page 3 lines 10-11).

Accordingly, reference no. 10 (PMID: 15589699) has been deleted and replaced with a new reference (PMID: 33368246) (page 3 lines 10-11).

Page 2, line 54. The authors provided a list of studies that, they claim, do not support the low vitamin D/dementia association. However, the list includes PMID: 26890771 which, although did not find association between vitamin D and incident dementia, showed that low vitamin D levels were associated with poorer executive function; the study PMID: 32080146 was only cross-sectional and did not investigate for intake of vitamin D supplements; PMID: 31827432 was clearly underpowered and with too short a follow-up. These references are therefore inappropriately cited.

According with the Reviewer, the introduction has been reformulated and the references inappropriately cited were removed (page 3 lines 32-34 and page 4 lines 1-8).

Methods

Page 2, lines 81-85. I have noticed a lack of detail regarding the recruitment of elders. How and when were the participants recruited into the study? Was it a random sample of all persons aged 64 or over, or a consecutive series? What was the participation rate?

As requested, in the new version of the “study design” paragraph, we have better detailed how and when the recruitment of older persons occurred (page 4 lines 29-31).   

Page 2, line 86-90. Since this is a study on dementia, more information should have been given about the psychometric scales adopted. For example, has the MMSE been adjusted for age and formal education according to PMID: 21284770? In addition, were the participants taking antidepressants? Could this have possibly led to interference with the accuracy of cognitive assessment, perhaps favouring lower depression scores?

An Italian version of the MMSE was administered to our subjects (PMID: 21284770) (page 5 lines 1-2).   

A number of subjects manifested depression, as indicated by the mean values of the geriatric depression scale (GDS) reported in Table 2 for the overall participants and in Table 3 for the subjects categorized on diagnosis. We described the results obtained in discussion (page 11 lines 12-15).

Anyway, the multivariate linear regression analysis was adjusted for the frailty index (FI) that included also the depression as an item for its computation.

Page 2, line 91. Why was the MMSE score threshold of 27 chosen to define the absence of dementia? While most studies usually adopt the ≤ 24 threshold for dementia, other studies adopt other thresholds (reviewed in PMID: 25740785).

We thank the Reviewer, there was a misunderstanding and in the new version we have replaced the typo in the text (page 5 line 15).   

Lines 93-94. The information provided by the authors about the study approval by the ethics committee, and the procedure for acquiring informed consent are undetailed and lacking documentary indications. This is even more serious since most participants were demented, for whom precise guidelines exist (see PMID: 33771131).

In the new version of the manuscript, we have provided the number of the study approval obtained by the ethics committee of the Fondazione IRCCS Ca’ Granda Ospedale Maggiore Policlinico (Milan, Italy) (REGE 2.0, protocol number 1696, 4th june 2021) (page 5 lines 20-21).

As we have better described in the paper, all the subjects involved in the study consented personally or, if a person is mentally incapable of making a decision, through their legal guardian to the use of their clinical and biological data for appropriately anonymized research purposes (page 5 lines 17-19)

Page 3, lines 96-100. The description the frailty index calculation lacks details. If questions were asked directly to the elders, how demented or severely depressed participants who did not fully understand the questionnaire were managed? In addition, the instrumental activities of daily living scale (Table 1) were used to calculate the fragility index. But given that in Latin culture males rarely do housework, or do laundry or cooking, how the resulting biased responses to these items have been corrected?

As suggested by the Reviewer, the description of the frailty index computation has been deeply detailed in the new version of the paper (page 5 lines 24-27, 31-34).

Moreover, in the text we have better described the multidimensional approach that was used to clinically characterize our subjects (page 4 lines 34-35 and page 5 lines 1-8) .

Regarding to the Lawton Instrumental Activities of Daily Living (IADL) reported in Table 2 and Table 3, it was differently calculated in men and women (PMID: 18367931) (page 5 lines 6-8).

Page 4, lines 118-120. Plasma vitamin D levels do not typically have a normal distribution but are somewhat skewed toward higher values (PMID: 1503066). How can the authors think of entering them into a linear regression model without prior transformation?

We thank the Reviewer for this observation. This is clearly an oversight. In the new version of the manuscript, we have log transformed the vitamin D values and we have run a new statistical analysis. The new results were reported in the text (page 8 lines 12-15).

Page 4, lines 129-130. If I understand correctly, education was not expressed as the number of years spent in school, but was categorized into five levels of unequal duration. It is not easy, in this case, to interpret the meaning of the corresponding regression coefficient.

In the new version of the paper, education was expressed as the number of years spent in school making the results easier to understand (Table 2 and Table 3).

Discussion

Page 6, lines 191-193. This sentence, at the beginning of the discussion is, word by word, identical to a sentence in the introduction (lines 45-48). While this is not a flaw in itself, it denotes, at least, poverty of language.

As suggested, the sentence at the beginning of the discussion was modified (page 10 lines 11-15).  

Minor remarks

The text is quite readable, but occasionally there are some misspellings or grammar problems, for instance:

- Abstract (line 17) and page 2, line 88. The correct term is "Hydrocephalus" not "Hydrocephalous".

- Line 21. Better “persons taking no supplements”.

- Introduction. Line 63, better “enrolled in the study” than “belonging to the study”. Line 65, please correct “lack of consensus”. Line 68, please correct “is still considered a debatable issue”. Line 109, better replace “lacking levels” into “deficiency”.

As suggested by the Reviewer, the misspellings have been corrected and the text was grammatically revised.  

Round 2

Reviewer 3 Report

The authors have made substantial changes to the manuscript, taking into account my observations. Some dated references have been appropriately replaced with newer ones. Therefore, now the study appears more solid and, to some extent, original. I have no further concerns other than in line 25 of the new abstract I would have preferred “most severe” rather than “deepest”, but it is a matter of personal preferences.

Author Response

In the reply letter we have discussed the comments of Editor and Reviewer 3
